# N-Glycosylation as a Key Requirement for the Positive Interaction of Integrin and uPAR in Glioblastoma

**DOI:** 10.3390/ijms26115310

**Published:** 2025-05-31

**Authors:** Gretel Magalí Ferreira, Hector Adrian Cuello, Aylen Camila Nogueira, Jeremias Omar Castillo, Selene Rojo, Cynthia Antonella Gulino, Valeria Inés Segatori, Mariano Rolando Gabri

**Affiliations:** 1Centro de Oncología Molecular y Traslacional, Universidad Nacional de Quilmes, Bernal B1876BXD, Argentina; maguiferreira2@gmail.com (G.M.F.); hectorcuello@live.com.ar (H.A.C.); aylen.nogueira@gmail.com (A.C.N.); jerecastillo13@gmail.com (J.O.C.); selenerojo25@gmail.com (S.R.); cynthia.gulino@gmail.com (C.A.G.); valeriasegatori@gmail.com (V.I.S.); 2Department of Cellular and Molecular Medicine, University of California San Diego, California, CA 92093, USA; 3Glycobiology Research and Training Center, University of California San Diego, California, CA 92093, USA; 4Laboratorio de Glicomedicina, Instituto de Biología y Medicina Experimental (IBYME), Buenos Aires C1428, Argentina; 5Consejo Nacional de Investigaciones Científicas y Técnicas (CONICET), Buenos Aires C1425FQB, Argentina

**Keywords:** IαV, uPAR, N-glycosylation, sialic acid

## Abstract

Integrin αV (IαV) and the urokinase-type plasminogen activator receptor (uPAR) are key mediators of tumor malignancy in Glioblastoma. This study aims to characterize IαV/uPAR interaction in GBM and investigate the role played by glycans in this scenario. Protein expression and interaction were confirmed via confocal microscopy and co-immunoprecipitation. The role of N-glycosylation was evaluated using Swainsonine (SW) and PNGase F. IαV glycoproteomic analysis was performed by mass spectrometry. Sialic acids and glycan structures in IαV/uPAR interaction were tested using neuraminidase A (NeuA) and lectin interference assays, respectively. Protein expression and their interaction were detected in GBM cells, but not in low-grade glioma cells, even in cells transfected to overexpress uPAR. SW, PNGase, and NeuA treatments significantly reduced IαV/uPAR interaction. Also, lectin interference assays indicated that β1-6 branched glycans play a crucial role in this interaction. Analysis of the IαV glycosylation profile revealed the presence of complex and hybrid N-glycans in GBM, while only oligomannose N-glycans were identified in low-grade glioma. N-glycosylation inhibition and sialic acid removal reduced AKT phosphorylation. Our findings demonstrate, for the first time, the interaction between IαV and uPAR in GBM cells, highlighting the essential role of N-glycosylation, particularly β1-6 branched glycans and sialic acids.

## 1. Introduction

Glioblastoma multiforme (GBM) is the most common and aggressive primary brain tumor affecting adults and accounts for 45.6% of all malignant primary brain tumors. The age-adjusted annual incidence of GBM increases with age, from 0.15 per 100,000 in children to a peak of 15.03 per 100,000 in patients aged 75–84 years. Regrettably, current therapies do not yield satisfactory outcomes. On average, GBM patients survive for only 12–18 months. However, 25% of these individuals live beyond a year, and 5% survive for more than five years [1]. A better understanding of the biology of GBM is crucial for unraveling its complexities and discovering new therapeutic targets.

Integrins are heterodimeric proteins composed of both the α and β subunits. In vertebrates, 18 different α and 8 different β subunits have been described, allowing the assembly and expression of 24 distinct heterodimers [2]. Integrin–ligand pairings can be categorized into four primary classes, determined by the type of molecular interaction involved. In particular, all αV-containing integrins together with two β1 integrins, as well as αIIbβ3, have been described as RGD receptors since they recognize short peptide sequences present in several extracellular matrix (ECM) proteins, such as vitronectin, fibronectin, and fibrinogen [3]. The role of RGD receptor integrins lies primarily in the regulation of cell behavior in response to the extracellular microenvironment. Altered integrin signaling is known to promote the invasive behavior of tumor cells, influence the tumor microenvironment to facilitate angiogenesis [4], participate in the interaction between the ECM and tumor cells [5,6], promote cancer immune escape [7,8], regulate stem cell function [9,10], and participate in the organ-specific targeting of metastatic cancer cells [11]. The expression of the αVβ3 and αVβ5 heterodimer has been widely described to be associated with malignant features of GBM [12,13,14,15]. αVβ3 was the first integrin to be abundantly detected in high-grade brain tumors [16], with almost 60% positivity reported in GBM samples [17]. Extensive research has underscored its role in sustaining GBM’s high proliferation rate, enhancing migratory and invasive capabilities, and promoting angiogenesis [18,19]. Furthermore, integrin participation in tumor radio- and chemo-resistance has been reported [20,21].

The association between integrin and urokinase-type plasminogen activator receptor (uPAR) has been described in several tissues and in tumors, in which it modulates signaling and cell behavior. uPAR is a glycosylated single-chain protein GPI-anchored with a molecular weight ranging from 50 kDa to 60 kDa. It exerts control over the plasminogen activation system, an extracellular proteolytic cascade, through its interaction with the serine protease urokinase-type plasminogen activator (uPA) and its inactive zymogen form, pro-uPA [22]. In addition to regulating extracellular plasmin activity, uPAR is a signaling receptor. It has been reported that signaling through uPAR activates the Ras-mitogen-activated protein kinase (MAPK) pathway, the Tyr kinase focal adhesion kinase (FAK), and the Rho family small GTPase Rac. Furthermore, FAK signaling can activate PI3K-Akt/PKB [23,24,25,26], which participates in the modulation of cell motility, invasion, proliferation, and survival. Since uPAR lacks a transmembrane domain, it requires cooperation with transmembrane receptors to drive intracellular signaling. The expression of uPAR has been highlighted in several cancers as a key factor in malignant behavior, and uPAR has been proposed as a poor prognostic marker [27]. In particular, the overexpression of uPAR in GBM is involved in epithelial–mesenchymal transition, and its association with poor prognosis has been reported [28,29]. Although several membrane proteins have been identified as possible co-receptors of uPAR, substantial evidence points to integrins as the main and most significant co-receptors of uPAR signaling [30,31]. In this regard, numerous pieces of evidence confirm that the interaction between integrins and uPAR is established mainly between the β-propeller region of the integrin α chain and uPAR domain III [32,33].

Even though both are glycosylated proteins, very little evidence provides information on the involvement of glycans in this interaction [34]. Despite significant advances in understanding the role of glycans in several types of tumors, the impact of glycosylation and the potential of glycans as therapeutic targets still merit further investigation [35]. Aberrant glycosylation involves changes in the glycosylation patterns of normal cell progenitors and occurs alongside the multistep process of malignant transformation [36]. These altered glycosylation profiles are linked to cellular characteristics that facilitate tumor progression, including adhesion to the ECM, migration, invasion, inhibition of apoptosis, host immunoregulation, and resistance to chemotherapy [37].

In this study, we demonstrated the interaction between integrin αV (IαV) and uPAR in human GBM cells and identified a differential pattern of glycans in the low-grade counterpart. In addition, we present compelling evidence of the significant role played by N-glycosylation in this interaction.

## 2. Results

To begin to explore the interaction between uPAR and IαV, we first examined the expression of glycoproteins in A172 and LN229 human GBM cell lines and in SW1088 human glioma of low grade by WB. Our results showed that these proteins are expressed in both GBM cell lines, although in SW1088 uPAR expression could not be detected (Figure 1A). To evaluate the interaction between these two glycoproteins, we first performed CM image analysis. Pearson’s coefficient analysis revealed a high co-localization rate of these two glycoproteins in both GBM cell lines (Figure 1B). In the same line, Co-IP confirmed this interaction in both cell lines when IP was performed on IαV or uPAR (Figure 1C). Since uPAR was not detected in the SW1088 cell line, PLAUR gene transfection was conducted to induce uPAR expression in these cells, which were subsequently designated as uSW1088 (Figure 1A). Notably, although uPAR expression was confirmed in uSW1088 cells, no molecular interaction between this protein and IαV was detected (Figure 1B,C).

Given that both uPAR and IαV are glycosylated proteins with multiple N-glycosylation sites [38,39,40,41], we questioned to what extent this type of glycosylation is involved in their interaction. SW specifically inhibits alpha-mannosidase II, an essential enzyme for the maturation of N-linked glycoproteins [42,43,44]. PNGase is an enzyme that cleaves N-linked glycans from glycoproteins by targeting the glycosidic bond between the asparagine residue of the protein and the first N-acetylglucosamine of the glycan [42,43,45]. Figure 2A shows that both treatments were able to reduce N-glycosylation in the treated cells, as evidenced by the low staining observed by incubation with PHA-L. As shown by CM analysis, Pearson’s co-localization coefficient was nearly 50% lower for control cells than for SW- or PNGase-treated cells (Figure 2B). Similarly, when Co-IP assays were performed, no IαV/uPAR interaction was detected by WB in the treated cells (Figure 2C).

The amino acid sequence of IαV, stored in the UniProtKB/Swiss-Prot repository, predicted the existence of 12 potential N-glycosylation sites (UniProt ID: P06756). Upon MS analysis of the IαV obtained from the GBM cell line LN229, 6 of these 12 sites were found to be N-glycosylated (Appendix A). To map and characterize all the potential glycan structures of each specific glycosite, the m/z values obtained from the MS-based glycoproteomic strategy were used. The main glycan structures for each position are depicted in Figure 3A. The results showed that Asn704 contains only complex/hybrid N-glycans, while only oligomannose N-glycans are present at Asn945. In addition, at Asn74, Asn554, and Asn874, mostly complex/hybrid N-glycans were present, with a decreased abundance of core (Manα1-3(Manα1-6)Manβ1-4GlcNAcβ1–4GlcNAcβ1–Asn-X-Ser/Thr) or oligomannose structures. On the other hand, at Asn615, a higher percentage of oligomannose-type N-glycans was detected. With respect to the presence of sialic acid, only the glycan at Asn945 had no sialic acid present, whereas the glycan containing N-glycolylneuraminic acid (NeuGc), N-acetylneuraminic acid (NeuAc), or both were observed at all the other glycosites. The detection of NeuGc—a non-natural sialic acid in human cells—can be explained by the presence of this carbohydrate in the culture media, since its active incorporation by cancer cells from its microenvironment has been reported [46,47]. Conversely, the analysis of IαV glycosylation in the low-grade SW1088 cell line revealed glycan structures at only two glycosylation sites, Asn74 and Asn874, where only oligomannose structures were present. No sialic acid was detected in the SW1088 cell line. The relative expression levels given the microheterogeneity of the N-glycans associated with each glycosite of the protein are shown in Figure 3B.

To determine the participation of sialic acids in the interaction between IαV and uPAR, cultured cells were treated with NeuA, an enzyme that removes the terminal sialic acids of glycan structures. As shown in Figure 4, NeuA treatment decreased IαV/uPAR colocalization, as determined by CM and Co-IP assays in both GBM cell lines.

To further our analysis, the interference of glycan-mediated protein interactions was evaluated using lectins. The PHA-L lectin mainly binds to β1-6 branched N-glycans, which were detected at the N74, N554, and N874 glycosylation sites of IαV in the GBM cell line. ConA recognizes oligomannose-type N-glycans that are detected at the N945 glycosylation site in both glioma and GBM cell lines and at the N74, N554, N615, and N874 glycosylation sites in the IαV from the GBM cell line. As expected from the glycosylation profile of GBM cells, incubation with PHA-L lectin strongly inhibited the interaction between IαV and uPAR. However, no effect was observed after ConA incubation (Figure 5), indicating the participation of β1-6 branched N-glycans in the IαV/uPAR interaction.

Finally, to assess the impact of glycans on IαV/uPAR-associated signaling pathways, AKT phosphorylation was evaluated in response to SW and NeuA. Both treatments resulted in decreased AKT phosphorylation in the LN229 and A172 cell lines. Interestingly, the upregulation of uPAR expression in these cell lines by transfection of the PLAUR gene resulted in increased AKT phosphorylation in the cells named uLN229 and uA172, suggesting the active involvement of uPAR in this signaling pathway. Similarly, SW or NeuA treatment strongly inhibited AKT phosphorylation in these cells (Figure 6).

## 3. Discussion

The interaction between uPAR and integrins is highly relevant in the progression of many types of cancer, and these proteins are key players in processes such as cell migration, invasion and metastasis. uPAR, a glycosylphosphatidylinositol-anchored cell surface receptor, binds to uPA, activating proteolytic cascades that degrade the ECM and facilitating tumor cell invasion. In parallel, uPAR can trigger intracellular signaling, thereby modulating physiological processes such as wound healing, immune responses and stem cell mobilization, as well as pathological events, including inflammation and tumor progression [48,49]. Integrins and integrin-dependent processes play crucial roles in nearly every phase of cancer progression by actively participating in signaling pathways that promote cell survival, proliferation, and metastasis [50,51,52]. Both cell membrane receptors have been identified as key contributors to GBM malignancy.

Interactions between integrin and uPAR have been shown in several other tumors. For instance, in gastrointestinal cancers, the interaction between αVβ6 and uPAR is implicated in the regulation of downstream signaling following uPA binding [30,32] and in the induction of MMP9 secretion, thus facilitating the degradation of multiple ECM components [53,54]. In breast cancer, Annis et al. demonstrated the participation of integrin αVβ3 in tumor invasion via activation of SRC/MAPK signaling and FRA-1 phosphorylation [55].

Although some reports on GBM suggest an interaction between integrins and uPAR, no concrete evidence in this regard has been published. Among the few reports on this topic are the results published by Veeravalli et al., who demonstrated the downregulation of the expression of several integrins, such as α1, α2, α6, α7, α9, and αV, as well as β1 and β3, in response to the reduction in uPAR mRNA levels [56]. Their research underscores the critical roles of integrins, uPAR, and MMP9 in glioma tumor biology and proposes a possible interaction model between them [57].

Given that IαV and uPAR are relevant antigens associated with poor clinical prognosis in human GBM, we focused our efforts on understanding these kinds of interactions. By employing two high-grade GBM cell lines that express both proteins, we detected interaction between these proteins, as demonstrated by Co-IP assays and CM. Since no uPAR was detected in the low-grade glioma cell line, we attempted to induce its expression by transfecting the uPAR gene sequence, termed PLAUR. Interestingly, no IαV/uPAR interaction was observed in these cells, suggesting that the mere presence of the protein in low-grade glioma cells is not sufficient to establish a positive interaction. This finding suggested that an additional factor participates in the interaction between these two proteins.

Glycosylation plays a significant role in protein interactions, especially in cell membrane proteins, where a broad spectrum of glycan structures is observed. Aberrant glycosylation refers to the particular glycophenotype of cancer cells and involves modification of the glycosylation profile of normal cell progenitors and is concomitant with the multistage process of malignant transformation [36]. The glycophenotype of tumor cells may be based primarily on N- or O-glycans, depending on the biology of the tumor. While in neuroblastoma, the glycophenotype is mostly expressed based on O-glycans [58], the glycosylation profile found in GBM is primarily expressed by N-glycans [59]. An altered glycosylation profile has been associated with cellular features that promote tumor progression, such as adhesion to the ECM, migration, invasion, inhibition of apoptosis, host immunoregulation and resistance to chemotherapy [60].

The glycosylation of integrins has been demonstrated to be relevant for cell adhesion, migration, and survival in tumor cells [61,62,63]. There are more than 20 potential N-linked glycosylation sites on αβ integrin dimers [61]. Additionally, integrin αV specifically has 12 putative N-glycosylation sites. The presence of these N-glycan core structures has been demonstrated to be crucial for several reasons; they are essential for integrin heterodimerization, stabilization of the conformation, expression at the cell membrane, and interaction with ligands [64,65].

The structural association between integrins and uPAR has been extensively reported. Simon et al. [66] and Zhang et al. [67] reported that the surface loop of the β-propeller domain of the integrin α-chain in α3β1 and αMβ2 heterodimers functionally associates with uPAR. Subsequently, Chaurasia et al. further showed that integrin α5β1 interacts with the domain III region of uPAR [68]. Additionally, Ahn et al. suggested that interaction with uPAR requires the expression of the complete αβ heterodimer (e.g., αVβ6) rather than individual subunits [33]. Additional insights through docking simulations of integrin αVβ3 and uPAR showed that the β-propeller region of αV from αVβ3 interacts with domains II and III of uPAR [69].

Although the integrin/uPAR interaction has been well described, the role of glycans in this process has been scarcely evaluated. To the best of our knowledge, only one report has shown that treating melanoma cells with SW inhibits the interaction of αV or α3 with uPAR, suggesting an important role for N-glycosylation in this interaction [34]. Similarly, our results demonstrated that N-glycosylation plays a key role in the integrin αV/uPAR interaction since treatment with both SW and PNGase inhibited this interaction, as demonstrated by Co-IP and CM. Of the 12 reported potential glycosylation sites of IαV, six were found in the GBM cell line LN229 (N74, N554, N615, N704, N874, and N945) and only two were found in the low-grade glioma cell line SW1088 (N74 and N874). Interestingly, the glycan profiles of these cells showed substantial differences. The glycosylation profile of IαV obtained from the GBM cell line exhibited a high proportion of complex and hybrid glycans, with a limited presence of oligomannose-type glycans, except at position N945, which contained 100% oligomannose. In contrast, IαV from low-grade glioma glycosylation was characterized exclusively by oligomannose-type glycans at the two glycosylated positions. As mentioned before, the β-propeller domain of integrin is, according to the literature, involved in the interaction with uPAR, with N74 being the only glycosylation site within this domain. Although position N74 of IαV is glycosylated in both cell lines, the glycan profiles differ significantly; the GBM cell line predominantly contains complex-type glycans, while the low-grade glioma cell line primarily contains oligomannose-type glycans. To identify the participating glycan structures, we treated GBM cell lines A172 and LN229 with NeuA, which strongly affects the IαV/uPAR interaction, indicating the significant involvement of sialic acids, which are found at all glycosylated positions. Moreover, incubation with PHA-L, which binds to β1-6 branched N-glycans, also inhibited the IαV/uPAR interaction. Conversely, incubation with ConA, which recognizes oligomannose structures, does not inhibit the IαV/uPAR interaction. Our results suggest that β1-6 branched glycans could be present at positions N74, N554, and N874. The first site is located within the β-propeller domain, the second is adjacent to it, and the third is near the transmembrane domain, forming part of the Ig domain 3. Considering the role of the integrin β-propeller domain in contacting uPAR, the glycan at N554 is unlikely to participate in the interaction, and the involvement of N874 is even less expected. Therefore, we speculate that the predicted branched β1-6 glycan at N74 plays an important role in this interaction. However, further studies should be conducted to evaluate the participation of glycosite N554. In this sense, the analysis of point mutations in the glycosylation sites would be a remarkable asset to identify with greater certainty which position is involved in the evaluated interaction. Sato et al. carried out a series of experiments that demonstrated the involvement of N-glycosylation sites present in the β-propeller as critical players in the interaction between α5 and β1 by point mutation analysis [70]. More recently, the role of N-glycosylation of the β1 subunit in AKT signaling has also been demonstrated using this technique [71].

To assess the impact of integrin-related glycosylation on the cellular response, we evaluated the phosphorylation of AKT, a known Integrin/uPAR-derived signal transducer [72,73]. The treatment of GBM cell lines with SW inhibited AKT phosphorylation by more than 50%, indicating that N-glycans are involved in AKT-activating signaling. Similar results were observed after treatment with NeuA, suggesting that sialic acid is a relevant player in this process. We performed the same experiments in cells transfected with PLAUR, which showed an increase in the phosphorylation of AKT, most likely due to the overexpression of uPAR in both cell lines and its inhibition as a consequence of SW and NeuA treatment.

## 4. Materials and Methods

### 4.1. Cell Lines

The human low-grade glioma cell line derived from a diffuse astrocytoma (grade II) SW1088 and the human high-grade glioma cell line (grade IV) A172, obtained from the American Type Culture Collection (ATCC, Manassas, VA, USA), were grown in Roswell Park Memorial Institute (RPMI) 1640 medium (Sigma-Aldrich, Burlington, MA, USA). The human GBM cell line (grade IV) LN229 was grown in Dulbecco’s modified Eagle’s medium (DMEM) (Sigma-Aldrich, MA, USA). The cell lines were supplemented with 10% fetal bovine serum (FBS) (Sigma-Aldrich, MA, USA) and 80 μg/mL gentamicin (Northia, CABA, Argentina). The cells were maintained at 37 °C in a humidified atmosphere containing 5% CO_2_, and routine subculture was performed using trypsin-EDTA solution (Thermo Fisher Scientific, Waltham, MA, USA) following standard procedures. Monthly screening for Mycoplasma contamination was conducted using 4′,6-diamino-2-phenylindole (DAPI) staining (Vectorlabs, Newark, CA, USA) and visualization under a fluorescence microscope.

### 4.2. Glycosylation Inhibition and Removal

The cells were cultured in their respective media and incubated with 100 nM swainsonine (SW) (Sigma-Aldrich, MA, USA), 5 UI/mL peptide-N-glycosidase F (PNGase) (New England Biolabs, Ipswich, MA, USA), or 1 UI/mL neuraminidase A (NeuA) (New England Biolabs, MA, USA) for 24 h. The control cells were treated with an equivalent volume of phosphate-buffered saline (PBS). Treatments do not affect the viability or the uPAR and IαV protein expression of the cells (Appendix A).

### 4.3. Lectin Interference Assay

Cell monolayers cultured without FBS were incubated with 20 μg/mL of the lectins Phytohemagglutinin-L (PHA-L) or Concanavalin A (ConA) (Vector Labs, MA, USA) for 1 h at 37 °C. After this incubation, protein co-immunoprecipitation and immunofluorescence were conducted.

### 4.4. Plasmid DNA Transfection

For overexpression of the uPAR protein, cells were grown to approximately 90% confluence and transfected with the uPAR plasmid (HG10925; Sino Biological, Wayne, PA, USA) using Lipofectamine 2000 reagent (Invitrogen, Carlsbad, MA, USA), according to the manufacturer’s instructions. Immunofluorescence or Western blot (WB) assays were also conducted 48 h post transfection.

### 4.5. Immunofluorescence

A total of 2 × 10^5^ (for A172) or 5 × 10^5^ (for LN229 or SW1088) cells were seeded on coverslips (JSHD, JS, Yancheng City, China) in 6-well plates. The cells were washed with PBS and fixed for 10 min with 4% formalin in PBS at room temperature. Following fixation, the cells were blocked for 30 min with 1% BSA in PBS. Subsequently, the samples were incubated overnight at 4 °C with primary antibodies against IαV (1:200; Abcam Cat# ab179475, RRID:AB_2716738). After washing with PBS, the cells were incubated with FITC-conjugated secondary antibodies (1:200; Abcam Cat# ab6717, RRID:AB_955238) in PBS for 1 h at room temperature. After three washes with PBS, the cells were incubated with primary antibodies against uPAR (1:50; (Santa Cruz Biotechnology Cat# sc-376494, RRID:AB_11150125) and Alexa 594-conjugated secondary antibodies (1:200; Thermo Fisher Scientific Cat# A-11005, RRID:AB_2534073), as previously described.

For PHA-L (Vector Labs, MA, USA) binding, 0.5 μg of the lectin was incubated overnight at 4 °C. After washing with PBS, the cells were incubated with 0.5 μg of streptavidin-FITC (Vector Labs, USA) in PBS for 1h at room temperature.

Finally, the coverslips were incubated with 0.1 μg/mL DAPI (Sigma-Aldrich, MA, USA) for 15 min at room temperature and mounted on glass slides with 80% glycerol in PBS.

### 4.6. Confocal Microscopy

Confocal microscopy (CM) was performed using an inverted Leica TCS SP8 spectral scanning confocal microscope (Leica Microsystem, Wetzlar, Germany) equipped with specific lasers for the fluorochromes Alexa Fluor-594 (exc: 561 nm), FITC (exc: 488 nm) and DAPI (exc: 405 nm). Observation was conducted with a 60X objective, and the images were captured by sequential scanning configuring different detection channels for each fluorochrome using LasX software (v3.7.4.23463 Leica Microsystem, WZ, Germany) to digitize images with a resolution of 1024 × 1024 pixels.

Co-localization analysis of proteins was performed using Pearson’s correlation coefficient analysis, which was implemented through the Fiji image processing package (RRID:SCR_002285) of ImageJ software (v1.54f; RRID:SCR_003070). Briefly, each pixel in the first image was paired with the corresponding pixel in the second image, creating an intensity values dataset resulting from the combination of both channels. The Pearson’s correlation coefficient values were interpreted as follows; a value of +1 indicated a perfect positive correlation, values greater than +0.5 denoted a positive interaction, 0 represented no correlation, and −1 indicated a perfect negative correlation.

### 4.7. Protein Co-Immunoprecipitation

Protein co-immunoprecipitation (Co-IP) was conducted using the PierceTM Direct IP Kit (Invitrogen, MA, USA) following the manufacturer’s protocol. Briefly, cells at 90% confluence were lysed with lysis buffer containing protease inhibitor cocktail (Sigma-Aldrich, MA, USA) at 4 °C and centrifuged at 13,000 × g for 10 min. The supernatant was incubated overnight at 4 °C on an orbital shaker with agarose columns containing 5 µg of anti-IαV (Abcam, CB, Cambridge, UK; Cat# ab179475; RRID:AB_2716738) or 4 µg of anti-uPAR (Ab221680, Abcam, CB, United Kingdom) antibodies. The proteins retained in the column were washed and eluted using an elution buffer to obtain the desired fraction. Immunoprecipitated proteins were analyzed by WB.

### 4.8. SDS-PAGE and Western Blot

Monolayers containing 5 × 10^5^ cells were lysed with RIPA buffer supplemented with protease and phosphatase inhibitors (Sigma-Aldrich, MA, USA). The protein lysates were clarified, and the protein concentrations were normalized using the Pierce™ BCA Protein Assay Kit (Thermo Fisher Scientific, MA, USA). Immunoprecipitated proteins or 30 µg of total cell lysates were subjected to electrophoresis on polyacrylamide gels, followed by semidry transfer to 0.45 μm polyvinylidene difluoride membranes (GE Healthcare, Chicago, IL, USA). The membranes were blocked with low-fat milk and then incubated overnight at 4 °C with primary antibodies against IαV (1:5000, Abcam Cat# ab179475, RRID:AB_2716738), uPAR (1:100, Santa Cruz Biotechnology Cat# sc-376494, RRID:AB_11150125), AKT (1:500, R and D Systems Cat# AF1775, RRID:AB_354982), phospho-AKT (pAKT) (1:500, PACO02670, Assay Genie, DUB, Dublin, Ireland), and β-tubulin (1:5000, BD Biosciences Cat# 556321, RRID:AB_396360). After washing, the membranes were incubated for an hour with an HRP-conjugated secondary antibody against rabbit (Bio-Rad Cat# 170-6515, RRID:AB_11125142) or mouse (Bio-Rad Cat# 170-6516, RRID:AB_11125547) immunoglobulins. Finally, the membranes were visualized using a bioluminescence kit (Bio-Lumina, PBL, BsAs, Argentina) and membrane images were captured using a C-DiGit^®^ Blot Scanner (LI-COR, Lincoln, NE, USA). The pAKT/AKT variation is expressed as the percentage change in the difference between the optical densities of pAKT and AKT in treated cells, relative to the difference obtained between these bands in untreated cells.

### 4.9. Protein Concentration

For mass spectrometry assays, immunoprecipitates were concentrated using Vivaspin^®^ 500 Centrifugal Concentrators (Sartorius AG, UGOE, Göttingen, Germany), following the manufacturer’s instructions. Immunoprecipitation products were centrifuged at 12,000× *g* for 2 h until a final volume of 50 μL was reached. The concentrated samples were subjected to N-glycan analysis by mass spectrometry.

### 4.10. Mass Spectrometry

Mass spectrometry (MS) was conducted on IαV from cell line cultures. After the electrophoretic run, the gels were fixed for 3 h in a solution containing 50% methanol and 2% phosphoric acid. Following three washes, the gels were incubated for an hour in a balancing solution composed of 33% methanol, 17% ammonium sulfate, and 3% phosphoric acid. Next, Coomassie Brilliant Blue G 250 (Sigma Aldrich, MA, USA) was added at a final concentration of 0.06%, and the gels were incubated overnight. After three additional washes with bidistilled water under agitation, the bands corresponding to the proteins of interest were excised.

Protein digestion and MS analysis were performed at the Proteomics Service of CEQUIBIEM at the University of Buenos Aires/CONICET. In brief, the excised bands were sequentially washed with 50 mM ammonium bicarbonate (AB), 25 mM AB, 50% acetonitrile (ACN), and 100% ACN. The products were subsequently reduced and alkylated with 10 mM dithiothreitol and 20 mM iodoacetamide. In-gel digestion with 100 ng of trypsin (Promega, Madison, WI, USA) in 25 mM AB was carried out overnight at 37 °C. The resulting peptides were recovered by elution with 50% ACN/0.5% trifluoroacetic acid and concentrated by rapid vacuum drying. Finally, the samples were resuspended in 15 µL of water containing 0.1% formic acid. The peptide digestion products were analyzed by nanoLC-MS/MS using a nanoHPLC EASY-nLC 1000 (Thermo Fisher Scientific, USA) coupled to a QExactive mass spectrometer. A C18 precolumn (Acclaim PepMap 3 μm, 100 Å, 75 μm × 20 mm) was used for peptide desalination, followed by a 75 min gradient of H2O:ACN at a flow rate of 33 nL/min with an Easy Spray C18 column (2 mm × 150 mm). The data-dependent MS2 method was used to fragment the most intense peaks of each cycle. The raw MS data were processed using Proteome Discoverer software (RRID:SCR_014477), version 2.1.1.21 (Thermo Fisher Scientific, MA, USA). Fragmentation of a glycopeptide precursor ([M + 2H]^+^) clearly showed the decomposition of the oligosaccharide portion permitting the assignment of the peptide moiety (*m*/*z*) (Appendix A). Accordingly, as an example, in the N-glycopeptide ANTTQPGIVEGGQVLK that contains the glycosite N74 detected at 3484.52 ([M + 2H]^3+^) *m*/*z* peak could be assigned the glycan HexNAc(3)Hex(6)NeuAc(1). An incomplete precursor of this glycan was found at 3193.43 ([M + 2H]^3+^) m/z peak (highlighted). The glycan structures found were corroborated using the GlyConnect platform, and graphical representations were obtained from the GlyTouCan repository.

### 4.11. Statistical Analysis

Statistical analysis was performed using Prism 8 software (GraphPad Inc, San Diego, CA, USA, RRID:SCR_002798). The data are presented as the mean values ± standard deviations. Prior to statistical testing, the normality of the data was assessed. For comparisons involving two independent samples, the Mann–Whitney test was employed. Multiple comparisons between experimental groups were analyzed using ANOVA followed by Tukey’s post hoc test. Significance thresholds were set at * *p* < 0.05, ** *p* < 0.01, and *** *p* < 0.001.

## 5. Conclusions

Our findings demonstrate for the first time the interaction of IαV with uPAR in GBM cells and the major role of N-glycans, suggesting the essential participation of β1-6 branched N-glycans as well as sialic acids. Both structures are found at glycosylation position N74 of IαV within the β-propeller domain. This work provides novel and valuable insights into the interaction between IαV and uPAR, two significant proteins in GBM tumor biology, highlighting the crucial role of glycosylation.

## Figures and Tables

**Figure 1 ijms-26-05310-f001:**
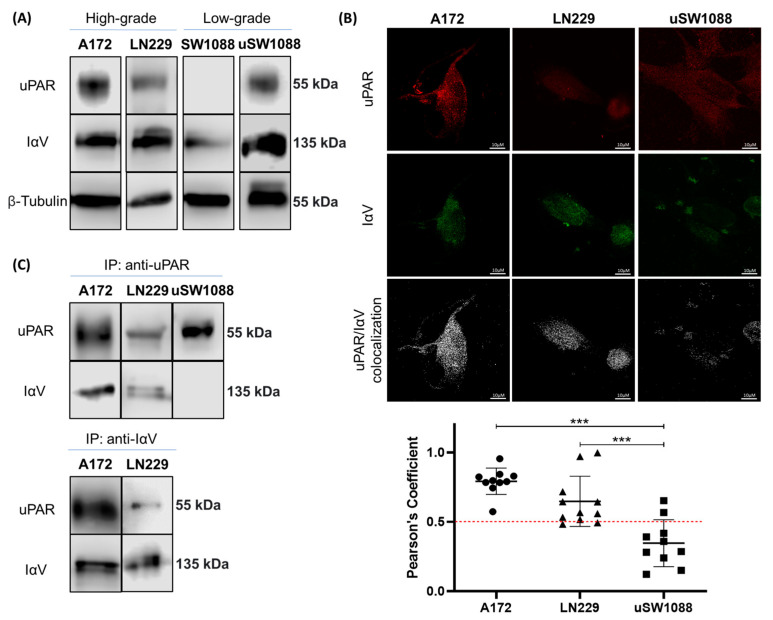
**Interaction between IαV and uPAR in human glioma cell lines**. (**A**) Representative WB analysis of IαV, uPAR, and β-tubulin expression in equal aliquots of whole-cell lysates from the A172, LN229, SW1088, and uSW1088 cell lines. (**B**) Protein co-localization analysis of CM from the A172, LN229, and uSW1088 cell lines. uPAR and IαV are shown in red and green, respectively, while co-localization is shown in white. Pearson’s coefficient analysis of a representative experiment of three independent datasets showing the means ± S.D. obtained from single cells of ten fields. (**C**) Evaluation of protein interactions through co-IP followed by WB. (*** *p* < 0.001, ANOVA followed by Tukey’s multiple comparisons test).

**Figure 2 ijms-26-05310-f002:**
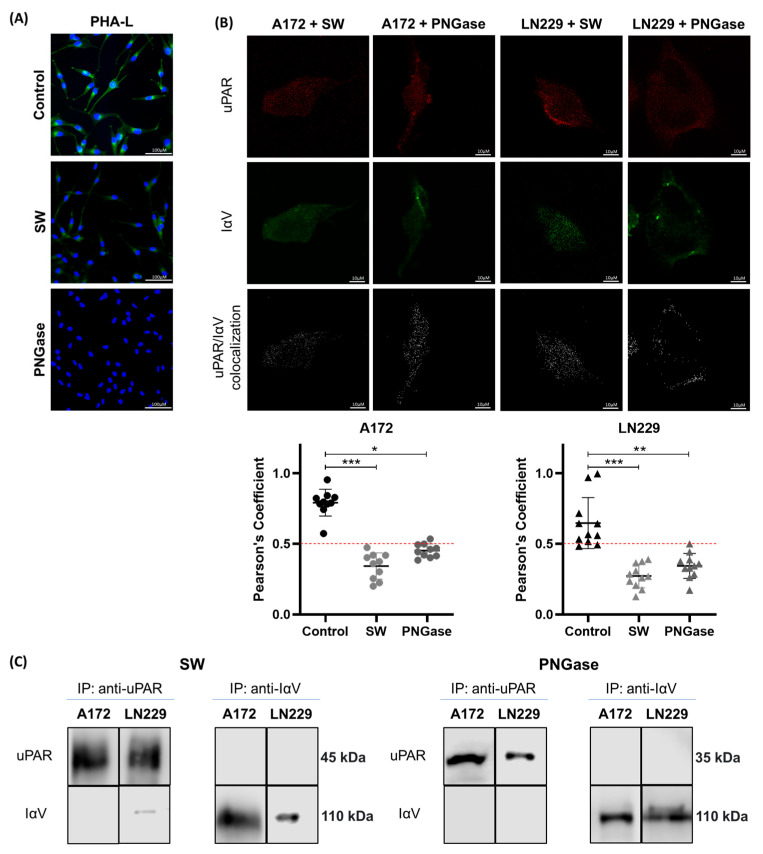
**Effect of N-glycosylation inhibition on the interaction between IαV and uPAR in the A172 and LN229 cell lines.** (**A**) PHA-L lectin binding after SW or PNGase treatment through immunofluorescence from the A172 cell line. PHA-L lectin and DAPI staining are shown in green and blue, respectively. (**B**) Protein co-localization analysis of CM after SW or PNGase treatments. uPAR and IαV are shown in red and green, respectively, while co-localization is shown in white. Pearson’s coefficient analysis of a representative experiment of three independent datasets showing the means ± S.D. obtained from single cells of ten fields (* *p* < 0.05, ** *p* < 0.01, *** *p* < 0.001, Mann–Whitney). (**C**) Co-IP of proteins followed by WB after treatment with SW or PNGase.

**Figure 3 ijms-26-05310-f003:**
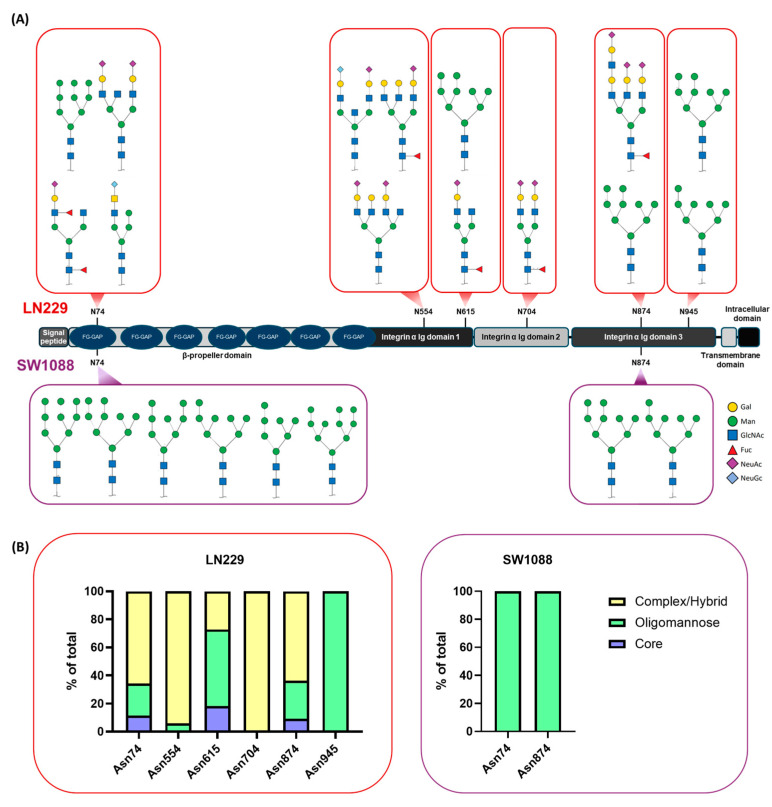
**Analysis of the N-glycans of IαV from low-grade tumors and GBM cell lines.** (**A**) Representation of the microheterogeneity of N-glycans on the IαV protein in the LN229 and SW1088 cell lines. (**B**) Relative abundance of core, oligomannose, and complex/hybrid N-glycans at each IαV position in LN229 or SW1088 cells.

**Figure 4 ijms-26-05310-f004:**
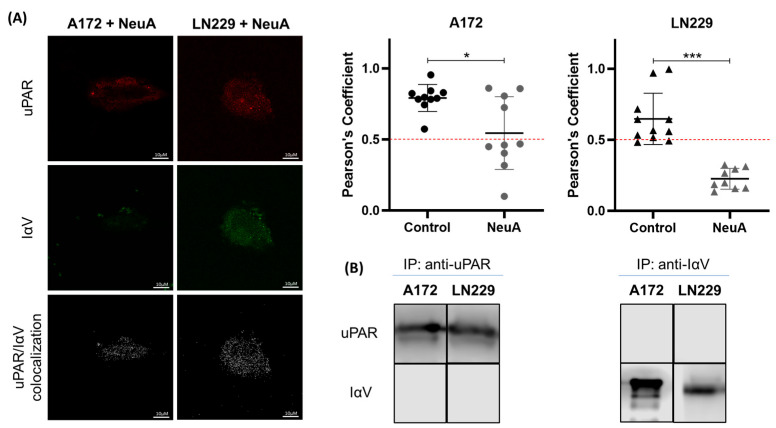
**Effect of sialylation on the interaction between IαV and uPAR.** (**A**) Representative protein co-localization analysis of CM after treatment with NeuA. uPAR and IαV are shown in red and green, respectively, while co-localization is shown in white. Pearson’s coefficient analysis of a representative experiment of three independent datasets showing the means ± S.D.s obtained from single cells of ten fields (* *p* < 0.05, *** *p* < 0.001). Mann–Whitney). (**B**) Co-IP of proteins followed by WB analysis after treatment with NeuA.

**Figure 5 ijms-26-05310-f005:**
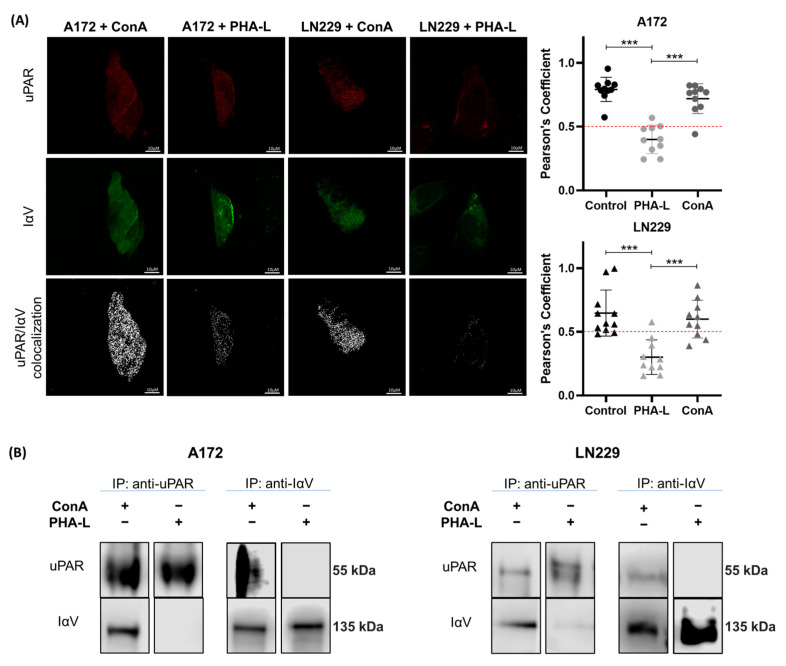
**Evaluation of the effect of lectin interference on the interaction between IαV and uPAR.** (**A**) Protein co-localization by CM after incubation with ConA or PHA-L. uPAR and IαV are shown in red and green, respectively, while co-localization is shown in white. Pearson’s coefficient analysis of a representative experiment of three independent datasets showing the means ± S.D obtained from single cells of ten fields (*** *p* < 0.001, ANOVA followed by Tukey’s multiple comparisons test). (**B**) Co-IP of proteins and WB after incubation with ConA or PHA-L.

**Figure 6 ijms-26-05310-f006:**
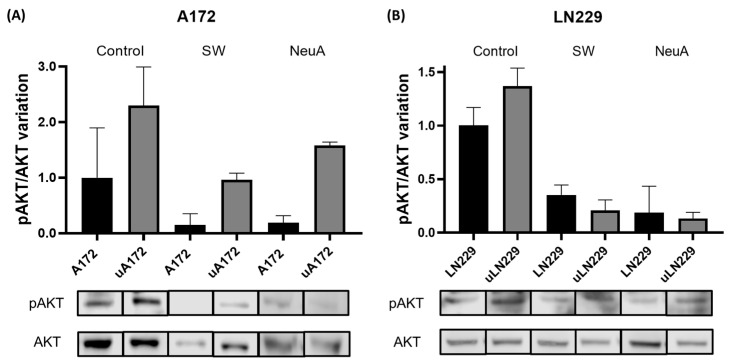
**Effect of glycosylation inhibition on AKT pathway activation.** Western blot showing pAKT and AKT expression and the pAKT/AKT variation after SW or NeuA treatment in (**A**) the A172 and uA172 cell lines or (**B**) the LN229 and uLN229 cell lines. The data are shown as a percentage of the control (WT).

## Data Availability

Data are contained within the article and Appendix A.

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
