# Peer review of "N-Glycosylation as a Key Requirement for the Positive Interaction of Integrin and uPAR in Glioblastoma"

_ijms, 2025, doi:10.3390/ijms26115310_

Round 1
Reviewer 1 Report
Comments and Suggestions for Authors
This manuscript investigates the interaction between integrin αV (IαV) and the urokinase-type plasminogen activator receptor (uPAR) in glioblastoma (GBM) cells, with a specific focus on the role of N-glycosylation in modulating this interaction. Overall, the study is rich in content, logically structured, and supported by convincing experimental data. However, I have several suggestions that may help improve the clarity and completeness of the study.
-
Stronger effect of sialic acid removal than total N-glycan removal?
Based on the results shown in Figures 2 and 4, treatment with neuraminidase A (NeuA)—which removes terminal sialic acids—appears to have a greater impact on IαV/uPAR colocalization than PNGase F, which removes the entire N-glycan. This observation raises the possibility that the presence of terminal sialic acids may be more critical than the entire N-glycan structure itself. The authors are encouraged to discuss this intriguing difference and propose potential mechanistic explanations.
-
Discrepancy between AKT phosphorylation and IαV/uPAR interaction levels
While PNGase F and NeuA treatments clearly reduce IαV/uPAR colocalization and interaction (Figures 2 and 4), the phosphorylation ratio of AKT in Figure 6 paradoxically appears elevated compared to the untreated control. This suggests that other compensatory signaling mechanisms might be involved. The authors should address this apparent inconsistency and provide possible interpretations.
-
Missing error bars in Figure 6B
Figure 6B lacks error bars, in contrast to Figures 6A and 6C. This omission limits the reader’s ability to assess the reliability and significance of the presented data. It is recommended that the authors add standard deviations or standard errors to Figure 6B for consistency and statistical rigor.
-
Mutational validation of glycosylation sites
The study would be significantly strengthened by the inclusion of glycosite-specific mutagenesis experiments. Point mutations that eliminate individual glycosylation sites (particularly N74, N554, and N874) could provide direct evidence for the functional importance of these glycans in mediating IαV/uPAR interaction. The authors are encouraged to consider incorporating such experiments or, at minimum, discuss their potential value.
Author Response
Comment 1: Based on the results shown in Figures 2 and 4, treatment with neuraminidase A (NeuA)—which removes terminal sialic acids—appears to have a greater impact on IαV/uPAR colocalization than PNGase F, which removes the entire N-glycan. This observation raises the possibility that the presence of terminal sialic acids may be more critical than the entire N-glycan structure itself. The authors are encouraged to discuss this intriguing difference and propose potential mechanistic explanations.
Response 1:
Following the comment of the reviewer, we performed a statistical analysis comparing the data sets of Pearson's coefficients obtained on confocal images of PNGase- versus NeuA-treated cells. No statistical differences were observed after this analysis showing that there is no evidence that suggests a different impact of these treatments in the co-localization of IaV and uPar. After the analysis of datasets comprising control samples, PNGase-treated samples, and NeuA-treated samples, statistical evaluation using one-way ANOVA indicates that comparisons between control and treated groups yield statistically significant differences (with at least one p-value < 0.05). In contrast, the comparison between the two treatment groups (PNGase vs. NeuA) within the A172 and LN229 cell lines does not demonstrate statistical significance, with p-values of 0.4110 and 0.1225, respectively.
Comment 2: While PNGase F and NeuA treatments clearly reduce IαV/uPAR colocalization and interaction (Figures 2 and 4), the phosphorylation ratio of AKT in Figure 6 paradoxically appears elevated compared to the untreated control. This suggests that other compensatory signaling mechanisms might be involved. The authors should address this apparent inconsistency and provide possible interpretations.
Response 2:
The uploaded Fig 6 in the submission process, columns labeled “wt” refer to untransfected cell lines (A172 in Fig 6 A and LN229 in Fig 6 B). Untreated cells are labeled as “Control” and treatments as SW and NeuA. In this revision process, we updated the figure replacing “wt” by cell line names to make it clear. Our results show that untreated cells exhibit a higher pAKT/AKT variation than treated cells in all conditions tested.
Comment 3: Figure 6B lacks error bars, in contrast to Figures 6A and 6C. This omission limits the reader’s ability to assess the reliability and significance of the presented data. It is recommended that the authors add standard deviations or standard errors to Figure 6B for consistency and statistical rigor.
Response 3:
Error bars in Figure 6b was added as requested
Comment 4: The study would be significantly strengthened by the inclusion of glycosite-specific mutagenesis experiments. Point mutations that eliminate individual glycosylation sites (particularly N74, N554, and N874) could provide direct evidence for the functional importance of these glycans in mediating IαV/uPAR interaction. The authors are encouraged to consider incorporating such experiments or, at minimum, discuss their potential value.
Response 4:
We fully agree that conducting a set of experiments involving the selected elimination of specific glycosylation sites through genetic modification would significantly strengthen this line of research. In fact, we are currently pursuing this approach as part of a complementary study to the present manuscript. This ongoing work aims to evaluate the downstream signaling pathways and assess the impact of the IAV–uPAR interaction on tumor behavior. However, for the moment we do not have any results, and it is not possible to achieve this objective within the deadlines established for sending the reviewers' responses. Following the reviewer suggestion, we include a paragraph in the discussion on this topic.
Reviewer 2 Report
Comments and Suggestions for Authors
Garbi and co workers have demonstrated the interaction between IαV and uPAR in GBM cells, highlighting the essential role of N-glycosylation, particularly β1-6 branched glycans and sialic acids. They have used N-glycan maturation inhibitor swainsonine, N-glycanase PNGase F and neuraminidase A treatment and followed their results with both confocal imaging and IP to establish their hypothesis. Experiments are playfully designed and conclusive to support theirs results. I have some unavoidable and serious concern about the writing and some control experiment. I suggest some minor changes before accepted for publications.
- Page 3, lines 99-106: What is that in the introduction!! I believe that part was suppose to be deleted before submission, which they forgot to do. This clearly represents lack of seriousness and involvement of the authors.
- Page 4, results and discussion of immunofluorescence; If two different antibodies from different host was added and finally imaged, then what was the reason of their consecutive addition and justification of the washing steps in between the additions.
- Page 7, line 280; please include reference for uPAR and IaV ate already being reported as glycosylated.
- Page 7, line 282; Add references for swainsonine inhibition (Front. Mol. Bioscience., 2023, 10,1286690.; Nat. Chem. Biology., 2025, 21, 681-692.; Annual Review Biochemistry, 1987, 56, 497-534.
- Page 7, line 284; Add references for PNGaseF (Front. Mol. Bioscience., 2023, 10,1286690.; Nat. Chem. Biology., 2025, 21, 681-692.; Analytical Biochemistry, 1989, 180, 195-204)
- Page 7, line 284; "One it was......interaction" this sentence is incomplete and makes no sense. Please rewrite the sentence.
- Figure 2. The author should run control experiment with EndoH and mannosidase I inhibitor kifunensine to further rule out the involvement of oligomannose or hybrid N-glycans.
- Page 8, line 301; These is no Table S1
- Figure 3. I was wondering what is the difference between oligomannose and core N-glycan structure here. If core means chitobiose core only, please indicate that in the figure.
- Page 12, Figure legend B) for figure 6; What is presented here in the graph! Is it 'percentage' or fold change?
- Page 14. line 476; supplementary material Table S1. Again nothing available in the associated link.
- Finally all these full length blots mean nothing in the additional file provided if they are not properly labeled. Please either Lebel them properly or just remove them.
Comments on the Quality of English Language
Some sentences are not complete and impossible to follow. I have already pointed out couple of such instances in the previous comment section. Overall writing needs grammatical corrections for general audience.
Author Response
Comment 1: Page 3, lines 99-106: What is that in the introduction!! I believe that part was suppose to be deleted before submission, which they forgot to do. This clearly represents lack of seriousness and involvement of the authors.
Response 1:
Unfortunately, an earlier version of the final manuscript was erroneously uploaded to the submission. Reviewer 2 is correct in his comments about the presence of the last paragraph of the submitted introduction. We have deleted this paragraph and worked on improving the introduction.
Comment 2: Page 4, results and discussion of immunofluorescence; If two different antibodies from different host was added and finally imaged, then what was the reason of their consecutive addition and justification of the washing steps in between the additions.
Response 2:
The methodology employed in our experiments follows an optimized protocol for multicolor immunofluorescence staining aimed at minimizing background noise and ensuring reliable detection of the target protein. In accordance with the procedures outlined in the manufacturers' protocol manuals, the anti-antigen antibody incubations were performed in sequential steps, each separated by thorough washing steps, to achieve high-quality imaging results.
https://www.novusbio.com/support/support-by-application/multicolor-immunofluorescence-immunocytochemistry/protocol?srsltid=AfmBOooDM9MxwvoZzQ-TT1_p37nexM3t98NEdA0OWbmiYQvaXwuJcPNz
Comment 3, 4 and 5:
Page 7, line 280; please include reference for uPAR and IaV ate already being reported as glycosylated
Page 7, line 282; Add references for swainsonine inhibition (Front. Mol. Bioscience., 2023, 10,1286690.; Nat. Chem. Biology., 2025, 21, 681-692.; Annual Review Biochemistry, 1987, 56, 497-534.
Page 7, line 284; Add references for PNGaseF (Front. Mol. Bioscience., 2023, 10,1286690.; Nat. Chem. Biology., 2025, 21, 681-692.; Analytical Biochemistry, 1989, 180, 195-204)
Response 3, 4 and 5: Refererences have been added as requested
Comment 6: Page 7, line 284; "One it was......interaction" this sentence is incomplete and makes no sense. Please rewrite the sentence.
Response 6: The aforementioned sentence was corrected as requested
Comment 7: Figure 2. The author should run control experiment with EndoH and mannosidase I inhibitor kifunensine to further rule out the involvement of oligomannose or hybrid N-glycans.
Response 7: We greatly appreciate the reviewer's recommendation. EndoH and kifunensine treatments are good experimental approaches to consolidate the results obtained by PNGase treatment that we observed by IP and also by IF. Unfortunately, at the moment we do not have these reagents in our laboratory. In addition, the time required to acquire the reagents and then perform the experiments exceeds the time frame set for answering the review.
Comment 8 and 11:
Page 8, line 301; These is no Table S1
Page 14. line 476; supplementary material Table S1. Again nothing available in the associated link
Response 8 and 11:
Supplementary table I was uploaded in the submission process as requested. In this round 1 revision process we uploaded it again in the “supplementary files” section.
Comment 9: Figure 3. I was wondering what is the difference between oligomannose and core N-glycan structure here. If core means chitobiose core only, please indicate that in the figure.
Response 9:
“Core” labelled bars in Fig 3 referees to the definition of core for N-glycans as it is present in the literature: “All eukaryotic N-glycans share a common core sequence, Manα1-3(Manα1-6)Manβ1-4GlcNAcβ1–4GlcNAcβ1–Asn-X-Ser/Thr” (Essentials of Glycobiology; Varki et al. 4th edition; Chapter 9). To avoid confusion, we have added a clarification in the main manuscript text.
Comment 10: Page 12, Figure legend B) for figure 6; What is presented here in the graph! Is it 'percentage' or fold change?
Response 10:
The pAKT/AKT variation is expressed as the percentage change in the difference between the optical densities of pAKT and AKT in treated cells, relative to the difference obtained between these bands in untreated cells. In accordance with reviewer comments, we have added a final sentence under “SDS-PAGE and Western Blot” in M&M describing the mathematical processing of the optical densitometry quantification of the bands resulting in the pAKT/AKT variation value.
Comment 12: Finally all these full length blots mean nothing in the additional file provided if they are not properly labeled. Please either Lebel them properly or just remove them.
Response 12:
In the submission process, the website asked for the original, unprocessed images to be uploaded. It seems that we misunderstood what the intent of this step was, as the images were uploaded without any modifications. In the “Non-published Material” section, a set of the original blots has been uploaded labeling them as requested.
Round 2
Reviewer 1 Report
Comments and Suggestions for Authors
The authors have addressed the relevant concerns raised in the review and have included the standard deviation values in Figure 6B. Although glycosite-specific mutagenesis experiments could not be added due to the time constraints of the review process, the additional experiments provided are sufficient to support the conclusions presented in the manuscript.